# Gene Augmentation of CHM Using Non-Viral Episomal Vectors in Models of Choroideremia

**DOI:** 10.3390/ijms242015225

**Published:** 2023-10-16

**Authors:** Lyes Toualbi, Maria Toms, Patrick Vingadas Almeida, Richard Harbottle, Mariya Moosajee

**Affiliations:** 1Development, Ageing and Disease, UCL Institute of Ophthalmology, London EC1V 9EL, UK; 2Ocular Genomics and Therapeutics, The Francis Crick Institute, London NW1 1AT, UK; 3cDNA Vector Research, German Cancer Research Center (DKFZ), 69120 Heidelberg, Germany; p.almeida@dkfz-heidelberg.de (P.V.A.);; 4Department of Genetics, Moorfields Eye Hospital NHS Foundation Trust, London EC1V 2PD, UK

**Keywords:** choroideremia, inherited retinal disease, non-viral gene therapy, S/MAR

## Abstract

Choroideremia (CHM) is an X-linked chorioretinal dystrophy leading to progressive retinal degeneration that results in blindness by late adulthood. It is caused by mutations in the *CHM* gene encoding the Rab Escort Protein 1 (REP1), which plays a crucial role in the prenylation of Rab proteins ensuring correct intracellular trafficking. Gene augmentation is a promising therapeutic strategy, and there are several completed and ongoing clinical trials for treating CHM using adeno-associated virus (AAV) vectors. However, late-phase trials have failed to show significant functional improvements and have raised safety concerns about inflammatory events potentially caused by the use of viruses. Therefore, alternative non-viral therapies are desirable. Episomal scaffold/matrix attachment region (S/MAR)-based plasmid vectors were generated containing the human *CHM* coding sequence, a GFP reporter gene, and ubiquitous promoters (pS/MAR-CHM). The vectors were assessed in two choroideremia disease model systems: (1) *CHM* patient-derived fibroblasts and (2) *chm^ru848^* zebrafish, using Western blotting to detect REP1 protein expression and in vitro prenylation assays to assess the rescue of prenylation function. Retinal immunohistochemistry was used to investigate vector expression and photoreceptor morphology in injected zebrafish retinas. The pS/MAR-CHM vectors generated persistent REP1 expression in *CHM* patient fibroblasts and showed a significant rescue of prenylation function by 75%, indicating correction of the underlying biochemical defect associated with CHM. In addition, GFP and human REP1 expression were detected in zebrafish microinjected with the pS/MAR-CHM at the one-cell stage. Injected *chm^ru848^* zebrafish showed increased survival, prenylation function, and improved retinal photoreceptor morphology. Non-viral S/MAR vectors show promise as a potential gene-augmentation strategy without the use of immunogenic viral components, which could be applicable to many inherited retinal disease genes.

## 1. Introduction

Inherited retinal diseases (IRDs) encompass a large group of clinically and genetically heterogeneous diseases that collectively cause progressive retinal degeneration with resultant sight loss [1,2]. They affect approximately 1 in 3000 people, representing an important cause of severe visual loss in the human population. In the United Kingdom, they are the commonest cause of severe sight impairment (blindness) certification in working-age adults, yet for the vast majority of cases, there is no approved treatment. To date, over 400 IRD causative genes have been identified [3,4].

Choroideremia is an X-linked chorioretinal dystrophy (prevalence: 1 in 50,000–100,000) involving progressive degeneration of the photoreceptors, RPE, and choroid [5,6,7]. Affected male patients typically develop night blindness in childhood, followed by restriction of the peripheral visual field, and finally, a decrease in central visual acuity, often leading to complete blindness later in life. Although female carriers often remain asymptomatic, it has been found that they show a spectrum of disease severity, with the more severe cases showing widespread chorioretinal atrophy similar to that observed in males [8,9]. Choroideremia is caused by mutations in the *CHM* gene, which encodes Rab escort protein 1 (REP1), a ubiquitously expressed 653-amino acid protein that forms an essential component of the catalytic Rab geranyl-geranyl transferase (RGGTase) II complex [10,11]. REP1 is involved in the intracellular trafficking and secretion of proteins and organelles by performing a post-translation lipid modification (prenylation) to Rab proteins, which are small GTP-binding proteins that act as key regulators of intracellular trafficking [10,11]. 

Gene-augmentation therapy is currently one of those most promising treatment options for choroideremia and other IRDs; this method conventionally involves the use of viral particles to package and deliver wild-type cDNA to the retina via intra-ocular injection, restoring expression of the defective gene [12,13]. One adeno-associated virus (AAV)-based gene therapy known as voretigene neparvovec already has approval by the Food and Drug Administration (FDA) and European Medicines Agency (EMA) for treating patients with biallelic *RPE65*-retinopathy [14,15]. Choroideremia is a good candidate for gene augmentation as it is a monogenic disease caused by loss-of-function mutations, with no known dominant negative effect or genotype–phenotype correlation. In addition, *CHM* cDNA is relatively short, 1.9 kb in length, and so can be accommodated by AAV vectors. However, among several clinical trials for *CHM* AAV therapy, a phase I/II trial (NCT02341807) did not report differences in visual acuity between injected and un-injected eyes at 2 years post-surgery, and a phase III multicentre study (NCT03496012) failed to meet the primary endpoints and key secondary endpoints after 12 months post-treatment [5,16]. Furthermore, intraocular inflammation was reported in two CHM patients [17,18], which has also been noted in AAV studies for other IRD genes, including *RPE65* [19] and *CNGB3* [20]. More recently, voretigene neparvovec has been reported to cause RPE atrophy with consequent photoreceptor loss in and outside of the bleb area, raising concern for viral-based retinal gene therapy [21]. 

Considering the drawbacks associated with viral gene therapy, non-viral alternatives are of great interest for IRDs, with one potential strategy being the use of plasmid vectors with extensive cloning capacity that incorporate a DNA motif known as the scaffold/matrix attachment region (S/MAR). S/MARs are genomic sequences at which the chromatin anchors to the nuclear matrix proteins during interphase, a function thought to be involved in gene regulation [22,23]. When incorporated into plasmids, they promote episomal maintenance (preventing genome integration), mitotic stability, and protection against epigenetic silencing, producing persistent gene expression both in vitro and in vivo [24,25]. pS/MAR are indeed replicated and equally segregated during mitosis. The mechanism of mitotic stability of pS/MARs is not fully understood. It has been reported that mitotic stability is supported by a specific interaction of this vector with components of the nuclear matrix, such as hnRNP-U/SAF-A, Topoisomerase II, Lamin B1, SATB1, or Histone H1 [26]. In previous studies, subretinal injection of S/MAR-containing plasmids complexed into nanoparticles has produced long-term transgene expression and improved the retinal phenotype in mouse models of RPE65- and ABCA4-related retinal disease [27,28,29].

In the present study, we have assessed the use of non-viral S/MAR vectors carrying the human CHM coding sequence to produce functional human REP1 expression in CHM patient-derived cells and chm mutant zebrafish (*chm^ru848^*), demonstrating the therapeutic potential of S/MAR vectors as non-viral alternatives for retinal gene therapy.

## 2. Results

### 2.1. Generation of CHM S/MAR Vectors and Rescue of Patient Fibroblasts

We generated a CHM S/MAR plasmid toolbox by cloning the full-length human CHM coding sequence (1.9 kb) into plasmids containing a downstream S/MAR sequence and one of five promoters (CMV, CAG, EF1a, hPGK, and hCHMp) (see Figure 1). The vectors also contained the green fluorescent protein (GFP) coding sequence from the copepod species *Pontellina plumata* (also known as copGFP). After validating the vectors via Sanger sequencing, we transfected HEK-293 cells with each vector to demonstrate their ability to drive overexpression of REP1 protein. The CMV and CAG ubiquitous promoters produced the highest expression at 48 h post-transfection (Figure 1C), and the CAG vector was used for subsequent experiments in fibroblasts. 

CHM patient dermal fibroblasts harboring the c.126C>G, p.(Y42*) nonsense mutation were transfected with pS/MAR-CAG-CHM, showing restoration of REP1 protein expression at 11 days following GFP-positive FACS sorting to 112.6 ± 32.2% (*n* = 3) of wild-type levels (Figure 2A,B), compared to the loss of protein expression in the non-transfected CHM fibroblasts. REP1 expression was maintained in transfected cells after three passages and more than 4 weeks (35 days) post-FACS sorting at levels comparable to that of the wild type (61.9 ± 27.9%, *n* = 3).

REP1 plays an essential role in lipid modification (prenylation) of Rab proteins and facilitates their intracellular membrane transport trafficking by binding to the hydrophobic prenylation motifs at the C termini; when REP1 is absent, a population of unprenylated Rabs builds up in the cells. By rescuing REP1 expression in CHM fibroblasts, we would expect the level of unprenylated Rabs to decrease if exogenous REP1 is functional. Therefore, the prenylation function was investigated by measuring the pool of unprenylated Rabs in transfected and non-transfected cells using an in vitro assay. At 7 days post-electroporation, it was found that transfected CHM fibroblasts showed a 75 ± 11.7% (*p* < 0.05) decrease in unprenylated Rabs compared to non-transfected CHM cells (Figure 2C). Unprenylated Rabs were not detected in the wild-type fibroblasts.

### 2.2. Expression of Human REP1 in Chm^ru848^ Zebrafish Embryos

Wild-type (wt) and *chm^ru848^* zebrafish embryos were micro-injected at the one-cell stage with pS/MAR-CMV-CHM. GFP expression could be detected in the embryos from around 6 h post-injection. At 5 dpf, fluorescent imaging showed mosaic expression of GFP throughout the body of the wt and *chm^ru848^* larvae (Figure 3A–D). *chm^ru848^* homozygous mutant embryos displayed characteristic systemic defects previously described [20]), including pericardial and abdominal edema, shorter body length, an uninflated swim bladder, and a persistent yolk sac. The ocular morphological features include microphthalmia, irregular eye pigmentation, cataracts, and widespread retinal degeneration. Wholemount examination of injected and un-injected *chm^ru848^* larvae did not show notable differences between the phenotypes. However, analysis of survival showed a mild but significant increase in vector-injected *chm^ru848^* zebrafish survival to 7.1 ± 0.7 days compared to 5.9 ± 1.17 days in un-injected larvae (*n* = 43 and *n* = 21 in un-injected and injected larvae, respectively, *p* < 0.0001) (Figure 3E). Western blotting using a human-specific antibody (2F1) detected human REP1 protein expression only in the injected mutant larvae at 5 dpf (Figure 3F). To assess whether the human REP1 was biochemically functional in vivo, a prenylation assay was performed (Figure 3G). Injected *chm^ru848^* zebrafish demonstrated a 59.5 ± 24.3% (*p* < 0.05) rescue in Rab prenylation levels at 5 dpf. 

GFP expression could be detected in the retina of injected zebrafish larvae, predominantly in the photoreceptor cells. Immunostaining of the five dpf *chm^ru848^* retinal sections for cone and rod-specific markers (using PNA lectin and anti-rhodopsin, respectively) was performed (Figure 4). In the wild-type retina, a typical expression pattern of cone and rod outer segments in the outer retinal layers was detected, which was highly disrupted in the mutant retina, showing loss of photoreceptor cells and abnormal outer segment morphologies. Co-detection of GFP with the photoreceptor markers in the injected *chm^ru848^* retina confirmed photoreceptor-specific expression of the pS/MAR-CMV-CHM and showed improvement in photoreceptor organization and morphology, with individual outer segments more easily distinguished.

## 3. Discussion

Gene-augmentation therapies are a popular strategy under development for treating IRDs; however, data from clinical trials, including those aimed at treating choroideremia, have indicated that AAV vectors may not be a suitable gene-delivery method for some disorders. In the present study, we have investigated the ability of non-viral episomal S/MAR vectors to produce functional REP1 protein in models of choroideremia to explore their utility as a potentially safer gene-augmentation method.

Initially, we generated S/MAR-containing plasmid vectors carrying the human CHM coding sequence with several different promoters, which were transfected into HEK-293 cells and patient-derived dermal fibroblasts. In CHM patient fibroblasts that show loss of REP1, the pS/MAR-CAG-CHM vector restored protein expression to levels similar to the wild type. The vector generated persistent expression, which was maintained over several passages and still detected at 35 days post-cell sorting, indicating their mitotic stability. Previous studies have demonstrated the mitotic stability of the pS/MAR vectors in vitro and in vivo [24,25,30]; for instance, stable transgene expression and episomal persistence were observed up to 170 days in murine and human pluripotent stem cells [24]. In wild-type conditions, REP1 is involved in the lipid modification of Rab proteins and facilitates their intracellular membrane transport trafficking by binding to the hydrophobic prenylation motifs at the C termini [31,32]. It has previously been shown that this biochemical function is disrupted in patient fibroblast models and *chm^ru848^* zebrafish, as evidenced by the detection of unprenylated Rabs using an in vitro prenylation assay [33]. We found that the pS/MAR-CAG-CHM vector was able to substantially reduce the percentage of unprenylated Rabs in both transfected CHM fibroblasts and injected zebrafish mutants, indicating that functional REP1 is generated by the vector both in vitro and in vivo. Previously, up to a 42% reduction of unprenylated Rabs was achieved in CHM fibroblasts when treated with translational readthrough-inducing drugs (TRIDS) [33], compared to 75% using non-viral vectors in the present study, which highlights the potential efficacy of this form of therapy. For future investigations, iPSC-derived RPE could be assessed as a more clinically relevant disease model.

To investigate the pS/MAR-CMV-CHM activity in vivo, the vector was injected into the CHM mutant zebrafish *chm^ru848^*. The ocular and systemic phenotypes of these zebrafish have been described previously [34,35]. The systemic defects in these mutants are attributed to a lack of REP2 protein in zebrafish, leading to embryonic lethality [34]. The CMV promoter was able to produce variable but relatively broad expression in zebrafish tissues, including the photoreceptors, which persisted up to the latest timepoint examined in this study at 5 dpf. Partial rescue was evidenced by mildly increased survival in the injected mutants, and GFP-expressing photoreceptors showed improved morphology compared to un-injected zebrafish. As mentioned, despite the mosaic expression pattern, prenylation function was still significantly improved in the injected fish. However, the overall disease morphology of the mutant fish remained relatively unchanged; this limited rescue may be attributed to differences in human and zebrafish REP1 proteins and the patchy cell distribution of vector expression, which may be broadened with zebrafish-specific promoters.

Although promising, non-viral strategies for gene therapy still do not outperform the best AAV capsids in terms of transfection efficiency in photoreceptor and RPE cells, and DNA vectors typically require coupling with molecular vehicles to aid their retinal entry. Much progress has been made towards improving transfection rates in the past decade, with the use of plasmids packaged into nanoparticles such as CK30PEG in the *Rpe65*^−/−^ mouse model of Leber congenital amaurosis [27,28] or, more recently, the successful repeated administrations of ECO nanoparticles in the *Abca4*^−/−^ mouse model of Stargardt disease [29]. Nonetheless, further proof-of-concept studies in higher-order animals are desirable before translating these strategies to patients.

There are several ongoing and completed clinical trials for AAV gene therapy for choroideremia [5,36]. Thus far, these have yielded some disappointing results, with failures to produce significant improvements in visual acuity and meet primary endpoints, in addition to reports of intraocular inflammatory events [5]. On top of the vectors themselves, the surgical procedures required for subretinal injection can cause retinal stretching, likely contributing to inflammation and atrophy. Damaging the tissue can trigger inflammation by releasing intracellular proteins, the extracellular matrix, or non-protein molecules like ATP [37]. CHM patients are likely more susceptible to these inflammatory processes as their remaining retina is typically small and friable. Therefore, less-invasive routes of administration, such as intravitreal or suprachoroidal injection, would be beneficial for the delivery of non-viral gene therapies. Unlike the single administration used for AAV therapies, the repeated administration of less-immunogenic non-viral vectors, via a less invasive route, may be an effective method for safe, long-term treatment.

Alternatively to gene augmentation, other therapeutic strategies being pursued to treat choroideremia include translational readthrough-inducing drugs (TRIDs) and antisense oligonucleotide (AONs)-based splice correction. For TRIDs, studies using gentamicin, paromomycin [38], PTC124, or PTC-414 [33] to treat cellular models and zebrafish harboring *CHM* nonsense mutations have shown significant recovery of REP1 expression and prenylation activity. AONs were used to correct the deep-intronic c.315-4587 T>A mutation in *CHM* in patient-derived lymphoblast cells [39]. Although promising, therapies such as TRIDS and AONs do not apply to a large number of IRD patients due to their specificity to certain mutations; non-viral gene augmentation could address such issues as it has wider suitability for mutations, with the additional benefit of a large gene size carrying capacity.

In summary, we have shown proof-of-principle that non-viral episomal vectors can produce functional REP1 proteins in cellular and zebrafish models of choroideremia, showing rescue of the underlying biochemical defect in both. This has shown promise as a potential gene-augmentation strategy without the use of immunogenic viral components, which could apply to many IRD genes.

## 4. Materials and Methods

### 4.1. S/MAR Vector Generation

The In-Fusion HD Cloning Kit (Takara Bio, San Jose, CA, USA) was used to clone full human CHM coding sequence into S/MAR plasmid backbones provided by Dr Richard Harbottle (DKFZ). The CHM sequence was amplified via PCR using CloneAmp HiFi PCR Premix (Takara Bio). The primers were designed to introduce 15 bp homologous overhangs for efficient recombination. First, 1–3 μg of vector backbone was digested with 1–3 units of XhoI and BmgBI for 1 h at 37 °C. For the recombination reaction, 100 ng of the vector and 50 ng of the insert were combined with the In-Fusion mix and incubated at 50 °C for 15 min. Stellar competent cells (Takara Bio) were used for transformation. Sanger sequencing of the full insert was performed for quality control.

### 4.2. Cell Culture

Wild-type (WT) and CHMY42X patient-derived human dermal fibroblasts were obtained from skin biopsies, as described in the methods of [40]. The cells were cultured in DMEM high glucose (Gibco #41966029, Waltham, MA, USA) supplemented with 15% FBS (Gibco #10500064) and 1% Pen/Strep (Gibco #15140122). After reaching 80% confluency, the cells were passaged using TrypLE Express Enzyme (Gibco #12605028), and the media were changed twice a week. HEK-293 cells were cultured in the same conditions but with the cell culture medium supplemented with 10% FBS. Cells were maintained at 37 °C under 5% CO_2_/95% air atmosphere, 20% oxygen tension, and 80–85% humidity.

Transfection of human dermal fibroblasts and HEK-293 cells was carried out using the Neon Transfection System 100 μL kit (Neon Electroporation System #MPK10025, Waltham, MA, USA). Briefly, cells were dissociated using TrypLE Express (Gibco #12605028) and counted using a Countess II Automated cell counter. For each transfection, 1 million cells were resuspended in 100 μL buffer R and mixed with 5–10 µg of vector DNA and electroporated with the following parameters: 1650 V, 10 ms, 3 pulses for human dermal fibroblasts and 1450 V, 10 ms, and 2 pulses for HEK-293 cells. GFP expression was monitored via fluorescent microscopy. Media was changed 24 h after electroporation.

### 4.3. Zebrafish Husbandry and Microinjection

Adult zebrafish (wild-type, AB-strain [wt] and *chm^ru848^*, (RRID:ZFIN_ZDB-ALT-040107-2) were bred and maintained at the UCL Bloomsbury campus zebrafish facility. *chm^ru848^* embryos were generated by matings of heterozygous zebrafish and raised at 28.5 °C in E3 medium. S/MAR DNA was injected into zebrafish embryos at the one-cell stage using a Picospritzer III microinjector. Approximately 25 ng of DNA was injected directly into the cell.

### 4.4. Western Blot

Cell samples or whole zebrafish larvae (10 per sample) were snap-frozen using dry ice. Samples were analyzed via Western blot assay, as previously described [21], using anti-REP1 (2F1 clone, Millipore #MABN52, RRID:AB_10808665, Burlington, MA, USA) primary antibody diluted 1:1000 followed by secondary anti-mouse IgG HRP conjugate diluted 1:10,000 (Sigma, St. Louis, MO, USA) in blocking solution (5% dry milk, PBS/0.1% Tween [PBS-T]). The membrane was stripped and re-probed with 1:5000 anti-β-actin antibody (Sigma-Aldrich #A2228, RRID:AB_476697, St. Louis, MO, USA) or anti-vinculin (Santa Cruz Biotechnology #sc-25336, RRID:AB_628438) as a loading control. Three independent experiments were performed to determine the mean protein expression in fibroblast samples.

### 4.5. Prenylation Assay

The prenylation assays were carried out on whole zebrafish embryos (pools of 10) and patient fibroblasts, as previously described [33]. Briefly, cytosolic protein extracts were obtained from zebrafish and fibroblast samples, which were then subjected to an in vitro prenylation assay using 5 µM biotin-labeled geranyl pyrophosphate (B-GPP; Jena Bioscience, Jena, Germany) as a prenyl group donor, 0.5 µM recombinant REP1 (Jena Bioscience), 0.5 µM recombinant Rab geranylgeranyl transferase (RGGT; Jena Bioscience), and 20 µM GDP in prenylation/lysis buffer at 37  °C for 1 h. The prenylation reaction was stopped with 6× SDS loading buffer, boiled at 95 °C for 5 min, and biotin incorporation was analyzed using Western blot with HRP-conjugated streptavidin (1:3000; Jackson ImmunoResearch, West Grove, PA, USA) and either anti-β-actin antibody (Sigma-Aldrich #A2228, RRID:AB_476697) or anti-vinculin (Santa Cruz Biotechnology #sc-25336, RRID:AB_628438, Dallas, TX, USA) as a loading control. Detection was performed using the ChemiDoc MP Imaging system (Biorad, Hercules, CA, USA). The number of biotinylated Rab proteins was then quantified by scanning densitometry using ImageJ and expressed as a function of the β-actin/vinculin signal. To allow relative comparisons, the biotinylated Rab population in the non-transfected CHM cell or *chm^ru848^* zebrafish samples was set to 100%. Three independent experiments were performed to determine the mean prenylation function.

### 4.6. Retinal Immunostaining

Retinal cryosections were prepared from zebrafish larvae that were immunostained, were stained for rod photoreceptors using anti-rhodopsin 4D2 (1:200; Abcam #ab98887, RRID:AB_10696805), followed by secondary Alexa Fluor 647 antibody (Thermo Fisher, Waltham, MA, USA; 1:1000). Rhodamine-labeled peanut agglutinin (PNA) lectin (1:200; Vector Laboratories #RLK-2200, Newark, CA, USA, RRID:AB_2336701) was also added at the secondary antibody step for cone cell detection. The slides were imaged using a Leica LSM 710 upright confocal microscope.

### 4.7. Statistics

Data are shown as mean values ± standard deviation from n observations. The Shapiro–Wilk test was used to test for normal distribution. Student’s *t*-tests or Mann–Whitney U tests were used for single comparisons. *p* < 0.05 was accepted to indicate statistical significance (*). GraphPad Prism software 7.0 was used for statistical analysis.

### 4.8. Ethics

The study protocol adhered to the tenets of the Declaration of Helsinki and received approval from the NRES Committee London-Riverside Ethics Committee (REC12/LO/0489). Written informed consent was obtained from all participants prior to their inclusion in this study. Zebrafish were maintained according to institutional regulations for the care and use of laboratory animals under the UK Animals Scientific Procedures Act and UCL Animal Welfare and Ethical Review Body (Licence no. PPL PC916FDE7). All approved standard protocols followed the guidelines of the ARVO Statement for the Use of Animals in Ophthalmic and Vision Research Ethics.

## Figures and Tables

**Figure 1 ijms-24-15225-f001:**
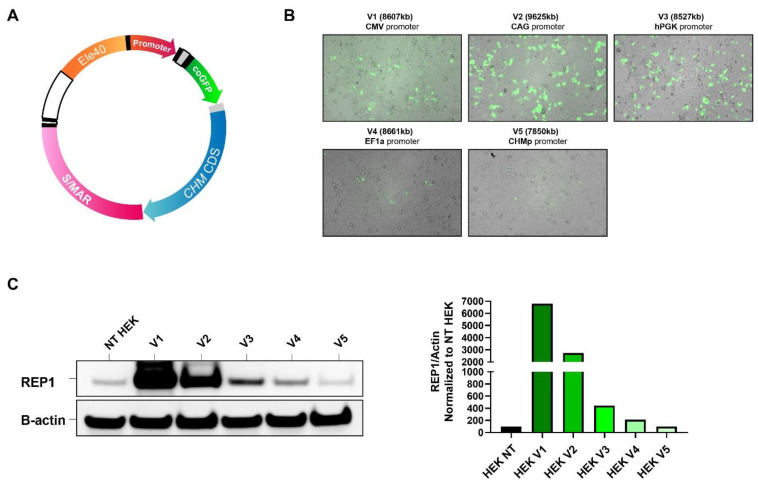
Generation of pS/MAR-CHM vectors. (**A**) pS/MAR-CHM vectors were generated by inserting the human CHM coding sequence (CDS) into the S/MAR plasmid backbone with 1 of 5 promoters: CMV (V1), CAG (V2), hPGK (V3), EF1α (V4), or CHMp (V5). The plasmids also contained a GFP sequence. (**B**) All 5 vectors produced GFP expression in transfected HEK-293 cells, which was observed at varying levels at 48 h post-transfection. (**C**) REP1 protein expression was examined by Western blot in the non-transfected (NT) and transfected HEK-293 cells, with the CMV (V1) and CAG (V2) promoter versions driving the highest expression levels.

**Figure 2 ijms-24-15225-f002:**
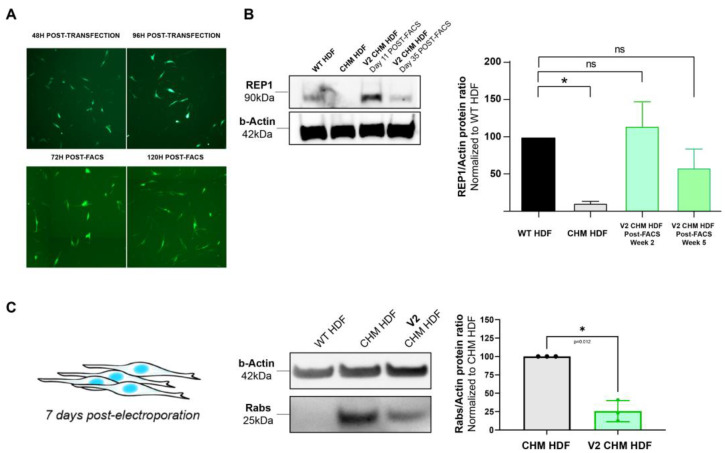
Rescue of CHM patient-derived fibroblasts. (**A**) Human dermal fibroblasts (HDF) from a CHM patient with the c.126C>G (p.Y42*) mutation were transfected with pS/MAR-CAG-CHM (V2). GFP expression was detected at 48 h post-transfection and was maintained at 120 h post-FACS. (**B**) Using Western blot, REP1 protein was detected in the transfected CHM patient fibroblasts at 11 days and 35 days post-FACS. REP1 was detected in wild-type control fibroblasts but was almost completely absent in non-transfected CHM fibroblasts. (**C**) As a measure of prenylation function, an in vitro prenylation assay was performed to detect the size of the unprenylated Rab protein pool in transfected versus non-transfected CHM fibroblasts at 7 days post-transfection. A significant decrease in the presence of unprenylated Rabs was observed in the transfected fibroblasts. * *p* < 0.05. (ns = non significant).

**Figure 3 ijms-24-15225-f003:**
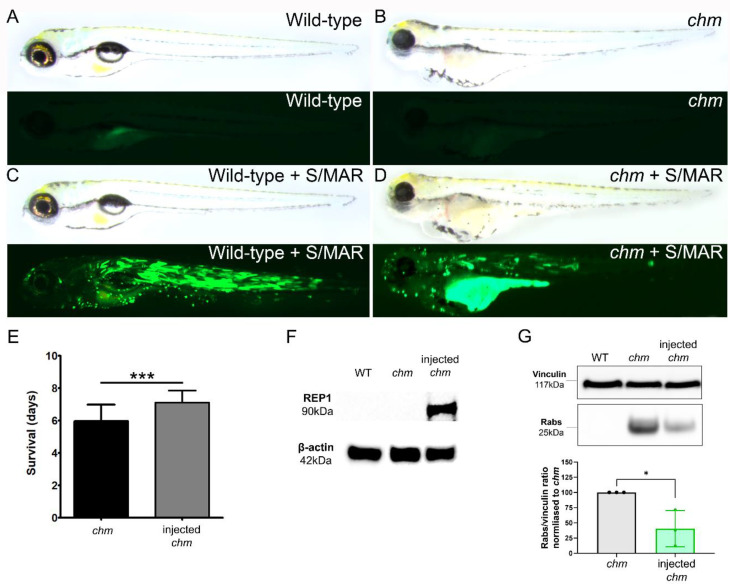
Micro-injection of *chm^ru848^* zebrafish with pS/MAR-CHM. Wholemount color and fluorescent images of (**A**) wild-type, (**B**) *chm^ru848^*, (**C**) pS/MAR-CMV-CHM-injected wild-type, and (**D**) pS/MAR-CMV-CHM-injected *chm^ru848^* zebrafish at 5 days post-fertilization. (**E**) Comparison of survival in injected and un-injected *chm^ru848^* zebrafish larvae. (**F**) Western blot for human-specific REP1 protein with the 2F1 antibody in injected and un-injected *chm^ru848^* zebrafish larvae. Human REP1 was detected in the injected zebrafish only. (**G**) Prenylation assay to detect levels of un-prenylated Rabs in wild-type, injected, and un-injected *chm^ru848^* zebrafish larvae (* = *p* < 0.05; *** = *p* < 0.001).

**Figure 4 ijms-24-15225-f004:**
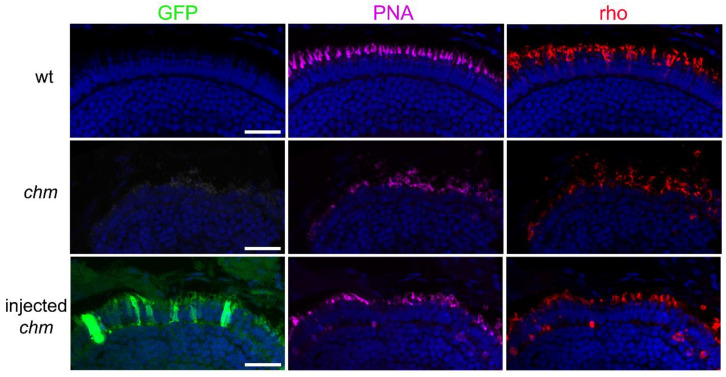
Retinal expression of pS/MAR-CHM in *chm^ru848^* zebrafish. Retinal sections from 5 days post-fertilization wild-type, *chm^ru848^*, and pS/MAR-CMV-CHM-injected *chm^ru848^* zebrafish were stained with PNA lectin and anti-rhodopsin to detect cone and rod cell outer segments, respectively. GFP was also detected in the photoreceptors of the injected zebrafish. Scale bar = 10 µm.

## Data Availability

The data presented in this study are available in the article.

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
