# Peer review of "Gene Augmentation of CHM Using Non-Viral Episomal Vectors in Models of Choroideremia"

_ijms, 2023, doi:10.3390/ijms242015225_

Round 1

Reviewer 1 Report

This is an excellent research paper showing some very promising results for the use of S/MAR containing plasmids as a possible future treatment option in choroideremia. It is very well writen and all the steps are documented.

I recommend it for publishing and I hope to see more results from this team, especially some proof that intraocular inflammation would hopefully be less of a problem with this approach. I am looking forward to a IRD therapy that may be repeatedly  administered by intravitreal or suprachoroidal injection.

Author Response

Thank you for your positive and supportive comments. We are moving forwards with these DNA vectors in more complex models, such as hiRPE and mouse models, with also a particular interest in ocular inflammation and innate immune response to the DNA and its carrier.

Reviewer 2 Report

IJMS 2023 CHM review

In this study Toualbi & Toms et al. investigates episomal scaffold/matrix attachment region (S/MAR) as a gene delivery vector for choroideraemia. Non-viral vectors are important and emerging alternative to AAV vectors. They primarily use fibroblast and one-stage zebrafish embryos to demonstrate effect in vitro and in vivo.

What is missing currently is the use in a relevant setting. If the aim is to use pS/MARs in gene therapy and provide an alternative to viral vectors, why don’t the authors perform a gene delivery experiment where retinal cells are targeted? Ideally side-by-side with an AAV vector. There are several experiments that can help to assess gene delivery to photoreceptors, such as using mice, organoids etc. The one stage embryo is not relevant from a therapeutic perspective. Based on the current data, we can not currently make any claims as to whether S/MARs can even be considered as gene delivery vectors to the retina.

Minor comments:

1) In general I disagree with the claim that choroideremia is a good candidate for gene augmentation. This is because it is unclear which cells to target. Every cell expresses CHM, however the disease only affects the eye. A probable reason for this is the lack of CHML expression (a similar protein) in certain cells in the eye.

2) what is the mechanism of mitotic stability of pS/MARs, if not integration? Is this piece of DNA also replicated?

Author Response

In this study Toualbi & Toms et al. investigates episomal scaffold/matrix attachment region (S/MAR) as a gene delivery vector for choroideremia. Non-viral vectors are important and emerging alternative to AAV vectors. They primarily use fibroblast and one-stage zebrafish embryos to demonstrate effect in vitro and in vivo.What is missing currently is the use in a relevant setting. If the aim is to use pS/MARs in gene therapy and provide an alternative to viral vectors, why don’t the authors perform a gene delivery experiment where retinal cells are targeted? Ideally side-by-side with an AAV vector. There are several experiments that can help to assess gene delivery to photoreceptors, such as using mice, organoids etc. The one stage embryo is not relevant from a therapeutic perspective. Based on the current data, we can not currently make any claims as to whether S/MARs can even be considered as gene delivery vectors to the retina.

Authors' answer. This work is the first proof-of-concept that these DNA vectors can partially restore REP1 protein expression and prenylation function. CHM S/MAR in this study demonstrated promising results in patient fibroblasts and zebrafish models. This study focuses more on the design of an efficient DNA vector more than the actual gene delivery and clinical translation. The next step of our research is to optimize non-viral carriers, nanoparticles, for these DNA vectors and to compare them with AAV, in more complex models including induced pluripotent stem cell derived RPE and retinal organoids, the mouse model and hopefully the pig model, which is still in development.

Minor comments:

1) In general, I disagree with the claim that choroideremia is a good candidate for gene augmentation. This is because it is unclear which cells to target. Every cell expresses CHM, however the disease only affects the eye. A probable reason for this is the lack of CHML expression (a similar protein) in certain cells in the eye.

Authors' answer. REP1, encoded by the CHM gene, is ubiquitously expressed but CHM is well characterised and we have recently shown that the choroid is also affected in addition to the RPE and photoreceptors (https://www.biorxiv.org/content/10.1101/2023.07.20.549875v1.full.pdf). Our group have also shown systemic features of CHM [1].

CHML (REP2) is expressed in the retina, and the choroideremia defect is considered to be underprenylation of Rabs, which preferentially bind to REP1. Replacing CHM should help restore the prenylation of Rabs and would help to slow the phenotype. More work to understand the fundamental disease processes is required but not the scope of this paper.

2) what is the mechanism of mitotic stability of pS/MARs, if not integration? Is this piece of DNA also replicated?

Authors' answer. pS/MAR are indeed replicated and equally segregated during mitosis. The mechanism of mitotic stability of pS/MARs is not fully understood. It has been reported that mitotic stability is supported by a specific interaction of this vector with components of the nuclear matrix such as hnRNP-U/SAF-A, Topoisomerase II, Lamin B1, SATB1 or Histone H1 [2,3]. A paragraph has been added to clarify in the introduction (89-93).

  1. Cunha DL, Richardson R, Tracey-White D, Abbouda A, Mitsios A, Horneffer-Van der Sluis V, et al. REP1 deficiency causes systemic dysfunction of lipid metabolism and oxidative stress in choroideremia. JCI Insight [Internet]. 2021 May 10 [cited 2022 Mar 23];6(9). Available from: https://pubmed.ncbi.nlm.nih.gov/33755601/
  2. Jenke BHC, Fetzer CP, Stehle IM, Jönsson F, Fackelmayer FO, Conradt H, et al. An episomally replicating vector binds to the nuclear matrix protein SAF-A in vivo. EMBO Rep [Internet]. 2002 Apr [cited 2019 Jun 13];3(4):349–54. Available from: http://www.ncbi.nlm.nih.gov/pubmed/11897664
  3. Stehle IM, Postberg J, Rupprecht S, Cremer T, Jackson DA, Lipps HJ. Establishment and mitotic stability of an extra-chromosomal mammalian replicon. BMC Cell Biol [Internet]. 2007 Aug 6 [cited 2018 Dec 20];8(1):33. Available from: http://bmccellbiol.biomedcentral.com/articles/10.1186/1471-2121-8-33

Round 2

Reviewer 2 Report

.